# A Narrative Review of Studies Comparing Efficacy and Safety of Citalopram with Atypical Antipsychotics for Agitation in Behavioral and Psychological Symptoms of Dementia (BPSD)

**DOI:** 10.3390/pharmacy10030061

**Published:** 2022-06-06

**Authors:** Haider Saddam Qasim, Maree Donna Simpson

**Affiliations:** Pharmacy Department, School of Medical Sciences and Dentistry, Orange Campus, Charles Sturt University, Orange, NSW 2800, Australia; masimpson@csu.edu.au

**Keywords:** BPSD, antipsychotics, citalopram, SSRIs, dementia, agitation

## Abstract

Background: Psychomotor agitation as part of the behavioral and psychological symptoms of dementia (BPSD) is one of the common issues found in aged care facilities. The current inadequate management strategies lead to poor functional and medical outcomes. Psychotropic interventions are the current preferred treatment method, but should these medications be the prescribers’ first preference? This review aims to compare pharmacological interventions for psychomotor agitation, judging them according to their effectuality and justifiability profiles. This is to be achieved by retrieving information from Randomized Control Trials (RCTs) and systematic reviews. Objectives: This review evaluates evidence from RCTs, systematic reviews, and meta-analyses of BPSD patients who have taken agitation treatments. Assessing the efficacy of citalopram, other selective serotonin reuptake inhibitors (SSRIs) and antipsychotic treatments were compared to each other for the purpose of improving agitation outcomes and lowering patient side effects. Methods: This review includes RCT that compared citalopram with one or more atypical antipsychotics or with a placebo, along with systematic reviews comparing citalopram (SSRI) with antipsychotics such as quetiapine, olanzapine, and risperidone. Studies were extracted by searching and accessing databases, such as PubMed, OVID, and Cochrane with restrictions of date from 2000 to 2021 and published in the English language. Conclusion: There are still a limited number of studies including SSRIs for the treatment of agitation in BPSD. SSRIs such as citalopram were associated with a reduction in the symptoms of agitation, and lower risk of adverse effects when compared to antipsychotics. Future studies are required to assess the long-term safety and efficacy of SSRI treatments for agitation in BPSD.

## 1. Introduction

A large portion of older people who are diagnosed with dementia experience one or more BPSD symptoms during their illness [1]. A recent systematic review stated that more than 80% of BPSD episodes manifest as agitation [2]. Psychomotor agitation and motor restlessness, characterized by inappropriate behavior [3], be it by verbal or vocal expression or physical activity that can be repetitive, aggressive, and often contradictory to social standards, are all symptoms of mood disorders [1,4]. Recent clinical guidelines recommended non-pharmacological interventions as the first attempt to manage agitation. However, if agitation causes distress and potentiates the risk of harm to others, then pharmacological approaches may be considered for alleviating agitation [1]. RCTs have evaluated different psychotropic interventions, such as antidepressants, antipsychotics, cholinesterase inhibitors, benzodiazepines, and anticonvulsants [5,6]. Most of these studies have made comparisons between different classes of psychotropics. A few RCTs have stated beneficial effects of citalopram for agitation associated with dementia [5,6,7], and suggested it as a safer alternative to antipsychotics. Citalopram consists of two enantiomers: R and S configurations, which are mirror images of each other. It has mild antihistamine properties that reside in the R enantiomer. Citalopram has an antidepressant action by potentiating serotonergic activity in the central nervous system and has minimal effects on noradrenaline and dopamine neuronal reuptake. It also has low affinity for muscarinic acetylcholine receptors and mild antagonist action at histamine H1 receptor. It has no significant effects on opioid mu receptors nor benzodiazepine receptors [8]. Citalopram is generally one of the better-tolerated SSRIs, and has favorable findings in the treatment of depression in the elderly; nevertheless, citalopram has an inconsistent therapeutic action at lower doses, and often requires dose increase to optimize the effect of treatment [8]. In the United States, citalopram is used off-label for alcohol use disorder, coronary arteriosclerosis, obsessive-compulsive disorder, panic disorders, premenstrual dysphoric disorder, and postmenopausal flushing [8].

There is, however, limited evidence from a direct RCT-double blind study comparing citalopram to other atypical antipsychotics in dementia. Therefore, this review aims to compare systematic reviews and RCTs to evaluate the effectiveness, safety, and acceptability of citalopram compared to other antipsychotics for psychomotor agitation in dementia. Citalopram is frequently used in older people, which has been proposed as an alternative to antipsychotics in the treatment of agitation in dementia [9]. Pollock et al. published three major datasets comparing citalopram with other psychotropic agents. The first open pilot study published in 1997 tried citalopram for managing behavioral disturbance of dementia [10]. The second study published in 2002 compared citalopram with perphenazine and placebo for the acute treatment of psychosis and behavioral disturbances in hospitalized-demented patients [11]. The third study published in 2007 presented data of a double-blind comparison of citalopram and risperidone for the treatment of BPSD [8]. All of these studies demonstrated the applicability of citalopram for managing agitation in dementia, but the generated data from these studies requires replication in a large double blinded randomized control trial (DB-RCT) and a placebo-controlled trial specific to a dementia population [11]. Based on the mentioned results of the previous studies conducted by Pollock et al. A study designed by Porteinsson et al. [6] included 186 participants who were randomized to receive cognitive and behavioral therapy (CBT) plus citalopram (initial dose 10 mg daily and titrated up to 30 mg daily over three weeks based on individual response and tolerability), with the other group receiving CBT plus placebo; both groups received these interventions regularly for the nine-week trial. The measures of Porteinsson’s study were based on: (1) Neuro-behavioral Rating Scale, (2) Agitation subscale (NBRS-A), (3) Modified Alzheimer Disease Cooperative Study-Clinical Global Impression of Change (Madcs-CGIC), (4) Cohen-Mansfield Agitation Inventory (CMAI), (5) Activities of daily living (ADLs), (6) Mini-Mental State Exam (MMSE), and (7) Comparing the adverse effects of citalopram at the level of 10 mg, 20 mg, and 30 mg [12]. The outcome of this study showed significant improvement with citalopram compared to placebo. The results of the scales and used of NBRS-A, mADCS-CGIC, CMAI, ADLs, and NPI showed significant improvement and use less of rescue benzodiazepines [13], but the NPI Agitation Scale gave less improvement, as this scale may be less sensitive than the others. However, the most frequent adverse effects of citalopram in this study were QT interval prolongation and worsening cognition points with increasing dose [9]. Cardiac events such as QT intervals, abnormal heart rhythm, and cardiac arrhythmias potentially occurred at doses over 40 mg daily, although there is no evidence of increased death among the dementia population taking over 40 mg [9]. 

### 1.1. Prevalence and Incidence of Dementia around the World

A recent global study on the burden of dementia, published by Lancet Public Health 2022, stated that the number of people with dementia is expected to increase. This study forecast the prevalence of dementia attributable to three risk factors, including the burden of dementia, injuries, and risk factors globally, by world region and at the country level. The outcome of this study estimated that age-standardized prevalence remains stable between 2019 to 2050 at a percentage change of 0.1% (95% uncertainty interval—7.5 to 10.8), whereas the number of individuals estimated to have dementia will be increased from 57.4 (50.4–65.1) million cases in 2019 to 152.8 (130.8–175.9) million case in 2050, with the highest number of increases in the Middle East and North Africa (367% (329–403)), Sub-Saharan Africa (357% (323–395)), while a lower incidence was reported in Western Europe (74% (58–90)) and in the Asia Pacific (53% (41–67)). Moreover, the estimation in the study stated that globally, there were more women with dementia than men (female to male ratio of 1.69 (1.64–1.73)); this pattern is forecast to continue to 2050. The implications of this study stated that a huge increase in the number of individuals affected by dementia can be expected by 2050 [14]. 

### 1.2. Phenomenology of Dementia

This term describes the cognitive and neuropsychiatric symptoms severe enough to interfere with the patient’s ability to perform their usual activities and causes a determinable decline from previous levels of functioning. These disorders include cognitive dysfunction, memory loss, reasoning impairment, and visual spatial impairment, along with language and communication issues; these symptoms are usually associated with symptoms of psychosis and agitation. Moreover, a recent qualitative study by Larsson et al. 2020 recognized four core additional symptoms of dementia (Alzheimer’s and Lewy body dementia): (1) wakefulness in attention and cognition abilities, (2) animated and recurrent visual hallucinations (3) rapid eye movement (REM) sleep behavior disorders leading to recurrent vivid dreams, and (4) parkinsonism, muscular slowness, rigidity, and increased risk and incidence of falls [15]. The phenomenology of dementia identifies the people, and further characterizes their experience of living with dementia by three major points: (1) disease impact, (2) self-perception and coping strategies, and (3) importance of others [15]. Regarding disease impact, cognitive complaints are the major issue under that heading, which is recognized by episodes of forgetfulness, difficulties remembering names, struggling to maintain conversation, being passive, slower speech and thoughts, and visuospatial problems. Most people with dementia demonstrated individualized complexities relating to their cognition and fluctuations. Fluctuations are characterized as being unable to sustain their current thoughts, losing their attention, and suddenly losing their consequences of thoughts, which could render a feeling of frustration. Moreover, dementia patients are suffering from significant restrictions to activity, and participation in social engagement/s, as many dementia sufferers experience deteriorating motor function with worsening gait and balance, which increases risk of falls and injuries. As a result, patients may develop subsequent feelings of fear around falling leading to the development of high-risk behaviors, such as moving slowly, using walking aids, and fears of leaving their house [16]. These behaviors contribute to the development of self-insufficiency, increasing isolation and reducing quality of life [16]. Regarding self-perception, dementia causes changes in both cognitive and physical aspects, and was found to negatively affect self-perceptions, as they felt the dementia influenced their skills, identity, cognitive functions (particularly in memory), physical and cognitive changes, being less accountable, and having fewer responsibilities. These issues cause many to feel like they are a burden to those around, based on their self-perception of their own illness and current abilities. These feelings of self-stigmatization are not always an accurate view of others’ perception [15]. 

### 1.3. Dementia Classification

There are four common types of dementia (full details in Appendix A).

Alzheimer’s disease: The most common type of dementia, making up 60% to 80% of dementia cases, often characterized by symptoms including; difficulty remembering recent conversations, names, and events, apathy, and depression, while the late stages symptoms usually progress to: impaired communication, disorientation, confusion, poor judgment, behavioral and psychological changes, difficulty of speech and swallowing, and decline in both fine and gross motor skills, including walking [17]. 

Vascular dementia: Vascular dementia is so named as it is the result of various combinations of chronic and acute cerebrovascular conditions and incidents that have a cumulative effect on cognition. About 5–10% of individuals with dementia showed evidence of vascular diseases [18]. 

Lewy body disease: about 5% of individuals with dementia showed evidence of only Lewy bodies (DLB), while many pathological autopsy cases reported Lewy bodies combined with Alzheimer’s disease changes [19] and the signs may occur including: uncoordinated movements, slow moving, tremors, and muscular rigidity (parkinsonism). 

Fronto-temporal lobar degeneration (FTLD): The age incidence of this kind of dementia occurs between 40 to 65 years, FTLD symptoms show marked changes in personality and behaviors and difficulty with producing and comprehending language, and memory [20]. This kind of dementia has features of primary progressive aphasia, Pick’s disease, progressive supranuclear palsy, and corticobasal degeneration [20]. There are several approved rating scales in dementia that are commonly used around the world, such as the Mini-Mental State Exam (MMSE) [21]; Neuropsychiatry Inventory (NPI) [22]; Montreal Cognitive Assessment [23]; Alzheimer’s disease Assessment Scale (ADAS) [24]; Behavioral Pathology in Alzheimer’s Disease [23]; Clinical Dementia Rating Scale [25]; Severe Impairment Battery [26]; Saint Louis University Mental Status Examination [27]; Functional Assessment Staging Test [28]; and Clinician’s Interview-Based Impression of Change [29]. All the listed scales are explained with details in Appendix A.

## 2. Materials and Methods

Studies were retrieved by searching electronic databases, including PubMed, OVID MEDLINE, and Cochrane Central Register of Controlled Trials for literature appurtenant to pharmacological interventions for treating agitation in dementia from 2000 until 2021. Search strategies are outlined in Appendix A. MeSH-indexed search terms ‘BPSD’, ‘dementia with agitation’, ‘psychomotor agitation’, ‘SSRI’, ‘citalopram’, ‘antipsychotics’, ‘olanzapine’, ‘haloperidol’, ‘risperidone’, and ‘quetiapine’. The results were filtered to include only RCTs, systematic reviews, and meta-analysis with an English language restriction. After that, the titles and abstracts were manually searched for those comparing SSRIs with antipsychotics. Moreover, the systematic reviews collected from the search were reported in accordance with the PRISMA. The collected data included the details of randomization, number of participants, dosage, duration of exposure, presence or absences, follow-up/loss of follow-up, adverse effects, and primary outcomes (See Figure 1).

Inclusion criteria of selecting clinical studies:Published in the English language;Randomized controlled trials RCTs, both experimental and non-experimental studies, non-RCTs, non-randomized (quasi-experimental) studies, observational, retrospective and prospective cohort studies, and analytical and cross-sectional studies;Published in peer-reviewed journals;Included any type of dementia and any level of severity in any setting;Focused on treatment and management of behavioral and psychological symptoms of dementia (included agitation, psychosis, and aggression);Included pharmacological interventions, compared different kinds of pharmacological interventions or compared to placebo;Included the outcomes of the pharmacological interventions or adverse effects.

Exclusion Criteria of selecting clinical studies: Any study focused on non-pharmacological interventions only;Studies focused on non-dementia population, such as health professionals or care givers;Low level of evidence, such as case reports, study protocols, commentaries, or design interventions;Population aged 17 years old or younger;Paper of other mental health or psychiatric disorders (not including dementia).

Inclusion Criteria of selecting guidelines:Were written in English;Focused on all types of dementia;Focused on the behavioral and psychological symptoms of dementia (BPSD);Presented recommendations in regard to agitation, psychosis, and aggressive behaviors;Were established in developed countries with reputable healthcare systems.

## 3. Results

Six RCTs were excluded as four of six RCTs compared only antipsychotics and two RCTs compared antipsychotics with benzodiazepines. Six systematic reviews were excluded, as four focused on antipsychotics and other antidepressants and two were comparisons of antipsychotics. Thus, the remaining three RCTs and two systematic reviews were included. 

### 3.1. Description of Systematic Reviews

Chaiyakunapruk et al. 2018 [30] designed a systematic review study. The result extracted 36 RCTs involving 5585 older people with dementia. Risperidone OR = 1.96 (95% CI 1.49–2.59) and SSRI (citalopram) OR = 1.61 (95% CI 1.02–2.53) were found to have significant effects compared to placebo. In addition, this review showed that haloperidol has less efficacy than other antipsychotics and SSRIs for controlling agitation [31]. The primary outcome of this review stated that risperidone and citalopram showed significant efficacy for agitation in dementia. Haloperidol and oxcarbazepine lack in efficacy and acceptability [7]. 

The Cochrane intervention review performed by Seitz et al., 2011 [32] intensively compared past studies on SSRIs used for agitation in dementia; included were five studies (citalopram, fluvoxamine, sertraline, and fluoxetine) using a placebo and four studies comparing SSRIs to antipsychotics (perphenazine vs. citalopram, haloperidol vs. sertraline, haloperidol vs. fluoxetine, and citalopram vs. risperidone). Overall, the outcomes stated that citalopram and sertraline were more effective than placebo and found no difference between SSRIs and antipsychotics in treatment of agitation [32].

### 3.2. Description of Randomized Controlled Trials

There are three recent RCT studies that have explored citalopram as treatment for agitation and aggression in dementia. 

In a six-month RCT [5], 75 participants at nursing homes were randomized to receive either citalopram, quetiapine, or olanzapine (n = 25 each group). Individuals included in the study met the criteria of NINCDS-ADRDA, clinically relevant agitation, and a history of psychotropic drugs before admission. The result of this trial showed that citalopram has similar efficacy to quetiapine and olanzapine for managing agitation: citalopram vs. quetiapine (OR 1 95% CI = 0.92, 1.7 at *p* = 0.935); citalopram vs. olanzapine (OR = 0.98, 95% CI 0.86, 1.2 at *p* = 0.849). Citalopram demonstrated lower all-case hospitalizations than quetiapine (OR = 0.92, 95% CI = 0.88, 0.95 at *p* = 0.016) and olanzapine (OR = 0.78, 95% CI = 0.64, 0.92 at *p* = 0.004). Moreover, citalopram also demonstrated a decreased occurrence of falls in comparison with olanzapine (OR = 0.81, 95% CI = 0.68, 0.97 at *p* = 0.012), but no difference was noted regarding instances of falls between citalopram and quetiapine. In addition, citalopram had the lowest incidence of orthostatic hypotension compared with quetiapine (OR = 0.8, 95% CI = 0.66, 0.95 at *p* = 0.032) and olanzapine (OR = 0.75, 95% CI = 0.69, 0.91 at *p* = 0.02). This trial revealed no differences observed regarding QT wave prolongation and infections [5]. 

A twelve-week randomized, double-blind controlled trial done by Pollock et al. was conducted at the University of Pittsburgh Medical Centre [7], comparing citalopram and risperidone for the treatment of psychosis and behavioral symptoms associated with dementia. In total, 103 participants were recruited; all met the criteria of at least one of the following moderate to severe target symptoms: agitation, aggression, delusion, and suspiciousness. Blinded randomization was in two groups, i.e., citalopram (n = 53) or risperidone (n = 50). Before and after mixed model analysis of the outcome measures was conducted using a neurobehavioral rating scale (NBR scale) and side effect rating scale (SER scale) at weekly and fortnightly intervals. The results of this trial showed agitation and psychosis symptoms decreased in both treatment groups: Neurobehavioral Rating Scale (NBRS) agitation score citalopram vs. risperidone (OR 0.11 95% CI = −0.28, 0.50 at *p* = 0.57); NBRS psychosis score (OR = 0.06, 95% CI = −0.35, 0.46 at *p* = 0.79); Udvalg for Kliniske Undersogelser side effect scale (UKU) total score (OR = 0.52, 95% CI 0.12, 0.91 at *p* = 0.01); UKU psychotic subscale score (OR = 0.55, 95% CI 0.15,0.94 at *p* = 0.007); UKU neurological subscale score (OR = 0.22, 95% CI = −0.17, 0.61 at *p* = 0.27). Additionally, the trial stated that there were significant side effects with risperidone but not with citalopram [7]. 

The CitAD randomized clinical trial is another study examining the effect of citalopram on agitation in Alzheimer’s dementia. The study was conducted by Porsteinsson et al. [6] as a randomized, parallel group, double-blinded, placebo-controlled trial that recruited 186 participants with Alzheimer’s dementia, clinically reported with agitation. The randomization was achieved by dividing 186 participants into two groups, i.e., 92 placebo group participants received psychosocial intervention and 94 intervention group participants received citalopram for 9 weeks. The intervention group received citalopram at 10 mg daily and gradually titrated to 30 mg daily over three weeks based on response and tolerability [6]. The results were established on the measures of neuropsychological rating scale agitation subscale (NBRS-A) and modified Alzheimer’s disease cooperative study-clinical global impression of change (mADCS-CGIC). Additionally, this study used other scales, such as CMAI, NPI, and ADLs for neuropsychiatry agitation and MMSE for mental status. The results of this trial showed significant improvement compared to the placebo group. The NBRS-A scale after 9 weeks shown OR −0.93 95% CI = −1.80, −0.06 at *p* = 0.04. The results of mADCS-CGIC shown 40% of the citalopram group showed improvements with OR 2.13 95% CI = 1.23, 3.69 at *p* = 0.01. Moreover, CMAI and NPI scales revealed significant improvements for citalopram group participants. However, QT intervals prolongation were seen at higher rates in the citalopram group than placebo (18.1 ms; 95% CI = 6.1, 30.1 at *p* = 0.004) [6]. As compiled below in Table 1.

## 4. Discussion

### 4.1. Quality of Evidence

#### Methodological Quality of the Systematic Reviews

The systematic reviews by Seitz et al. [32] and Chaiyakunapruk et al. [30] were of high quality, scoring 14/16 and 12/16, respectively, on the AMSTAR2 assessment tool (Table 2). Both reviews searched the PICO question and clearly stated the inclusion criteria. Both reviews expressed their methodology prior to conducting the articles, refining and both significantly justifying the protocols. The authors of both reviews explained their study designs, adequately justifying exclusions, and used comprehensive literature search strategies. 

The review by Seitz et al. applied the ‘risk of bias tool’ to assess bias in the results by methodological quality and summary graphs. Moreover, bias in this review was assessed by visual screening of funnel plots of the primary outcomes. The review by Chaiyakunapruk et al. examined the selected study bias by using the revised Cochrane risk of bias tool for randomization trials (RoB version 2). This tool measures deviation from intended interventions, bias in the measurements, bias due to results outcomes, and bias due to missed outcome data. In general, the risk of bias of the selected studies in this review was classified at low risk. The authors of both reviews provided a satisfactory explanation and a discussion of the results. Both reviews reported potential conflict of interest, but there is no information about the sources of funding they received for conducting the reviews. 

### 4.2. Methodological Quality of Most Recent Trials

The trial by Viscogliosi et al. [5] was conducted with 75 participants, selected after inclusion criteria were satisfied, including a diagnosis of Alzheimer’s dementia paired with clinically relevant agitation. The method used to generate the random allocation sequences was conducted by two experienced geriatricians who randomized the participants into three equal groups (n = 25 each group). However, there is no information on whether blocking or stratification was reported during randomization, and no information on whether any participants in the group were concealed until interventions were assigned. The authors did not specify what aspects of the trial were blinded, and if the persons administering the interventions or those investigating the outcomes had knowledge of what substance each patient was given. Regarding participant flow in this study, a diagram was absent, and there was a noted absence of information regarding what had happened to the groups for a period of 6 months during the trial. Additionally, there was a distinct lack of information regarding how many participants did not engage in follow-up. The strong point of this study has demonstrated the baseline demographic and clinical characteristics of inclusion and exclusion criteria. The numbers of participants in each group were compared and analyzed into two tables and graphs. The report also provides clear outcomes and a summary of results for each group [5]. 

The objective and the hypotheses of the trial by Pollock et al. [7] are well explained, comparing risperidone and citalopram for the treatment of agitation and psychosis in dementia. The hypotheses of this trial stated that citalopram is more effective for agitation, while risperidone proved more effective for psychosis. The period of the trial was 12 weeks, and randomized 103 dementia patients. The eligibility criteria for participants along with the settings and locations where the data were collected are mentioned in detail. The participants were randomized to either risperidone (n = 50) or citalopram (n = 53). The randomization was generated by a biostatistician at the beginning of the trial and the stratification of randomization might consider based on the presence or absence of the psychotic symptoms [7]. Only the research pharmacist had access to the information regarding the treatment assigned to each group, participants, and investigators. Only the assessors in this trial remained blind throughout the study. After randomization, the assigned participants were assessed after receiving citalopram or risperidone for three days, then seven days, then once a week for five weeks, then once fortnightly, and finally at the discharge from hospital. Additionally, the authors explained the details of intent-to-treat principles after randomization. In addition, the participants continued to administer the medications even after they were discharged under rigorous maintenance of the double-blinded conditions. The flow of participants through each stage of the trial was explained clearly in the methods and discussion, illustrated by a diagram. The author successfully demonstrated that each group reported the numbers of randomly assigned participants, intended treatments, and follow-up, completed and analyzed for the primary outcomes. The limitation of this trial is that it had no placebo, and it was established on the efficacy of citalopram and risperidone on evidence of only two previous trials [7].

Marano et al. conducted an RCT study that showed that citalopram reduced agitation when compared to placebo [6]. The objective of this trial was to assess the efficacy of citalopram for agitation in Alzheimer’s dementia. The study involved 186 patients in a double-blinded, randomized, placebo-controlled trial. The participants satisfied the inclusion criteria (>65 years, diagnosed with Alzheimer’s dementia, and reported with clinically significant agitation). The authors mentioned that the participants were randomized; however, there is no explanation of the method used to generate the random allocation sequence, and no details regarding whether the randomization exposed any restriction, blocking, or stratification [6]. It is mentioned in this study that the trial is a double-blinded treatment assignment with adherence to mask rating. However, there are no details about whether or not the participants in both intervention and control groups and those assessing the outcome were blinded, how blinding was achieved, and how blinding success was evaluated. A total of 186 participants were randomized to receive a psychosocial intervention, and either placebo or citalopram for a duration of 9 weeks. The initial dose is 10 mg for citalopram with gradual titration to a maximum of 30 mg daily over three weeks based on individual response. The flow of participants through each stage of the trial was mentioned in detail and illustrated by a diagram specifically for each group. This diagram also reported the numbers of randomly assigned participants, who received citalopram or placebo, steps of follow-up, and number of absences in the follow-up for each check-up visit. The primary outcome of this trial was based on scores from the neurobehavioral rating scale agitation subscale (NBRS-A) and mADCS-CGIC scale, CMAI scores, NPI, ADLs, MMSE, and adverse effects. In general, despite QT prolongation being reported in the citalopram group, the participants showed significant improvements of agitation compared to placebo [6].

## 5. Conclusions

Traditionally, atypical antipsychotics have been used as the primary method for treating episodes of agitation and psychosis for people with dementia. However, despite the effectiveness of these agents in managing the agitation, they still provide undesirable side effects and an increased risk of mortality. As a result, there are continued campaigns to reduce the use of antipsychotics in older people and promoting the use of alternative agents such as SSRIs. Although there is limited evidence collected in this review, the findings from the two systematic reviews and the three clinical trials indicated that SSRI antidepressants (citalopram) not only showed efficacy for treating BPSD, but were better tolerated compared to antipsychotics. This review also showed that citalopram reduces the symptoms of agitation when compared to placebo and it has fewer adverse effects compared with atypical antipsychotics, such as risperidone, quetiapine, and olanzapine. The outcome of this review suggests that further studies involving more participants from aged care facilities should be conducted, with a longer duration of the trials to assess the safety and the efficacy of citalopram for managing agitation in dementia in long-term use.

## Figures and Tables

**Figure 1 pharmacy-10-00061-f001:**
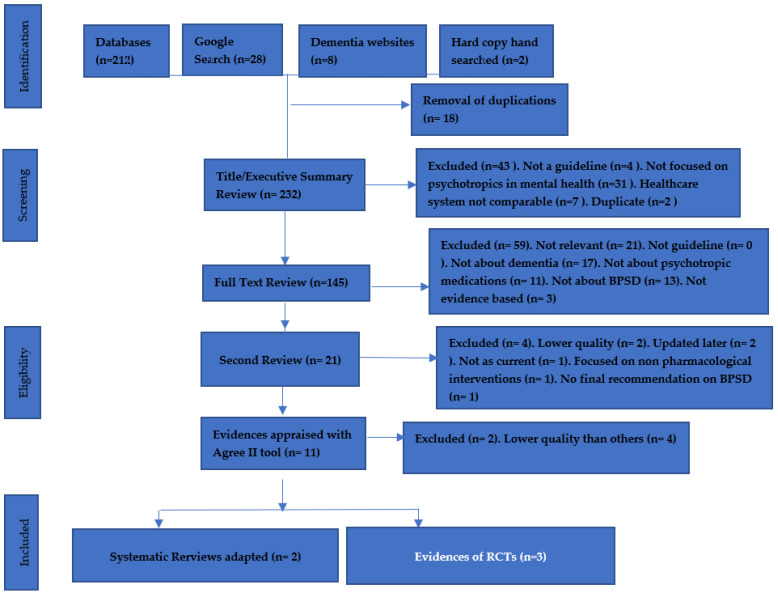
PRISMA framework for evidence review.

**Table 1 pharmacy-10-00061-t001:** Psychotropics-related outcomes of three randomized controlled trials.

Citation, Year	Number of Participants	Duration	Active Ingredient	Lost Follow-Up	Difference between Groups	Outcome
Viscogliosi et al., 2017 [5]	75 (citalopram n = 25), (olanzapine n = 25), (quetiapine n = 25)	26 weeks	Citalopram, olanzapine, quetiapine	Short duration of follow-up (no details)	Efficacy against agitation: citalopram vs. quetiapine (OR 1 95% CI = 0.92, 1.7 at *p* = 0.935), citalopram vs. olanzapine (OR = 0.98, 95% CI = 0.86, 1.2 at *p* = 0.849). Hospitalization: quetiapine (OR = 0.92, 95% CI = 0.88, 0.95 at *p* = 0.016), olanzapine (OR = 0.78, 95% CI= 0.64, 0.92 at *p* = 0.004). Occurrence of falls: olanzapine (OR =0.81, 95% CI = 0.68, 0.97 at *p* = 0.012). Incidence of orthostatic hypotension: quetiapine (OR = 0.8, 95% CI = 0.66, 0.95 at *p* = 0.032), olanzapine (OR = 0.75, 95% CI = 0.69, 0.91 at *p* = 0.02).	Citalopram has similar efficacy to quetiapine and olanzapine. Citalopram showed less all-case hospitalizations than both quetiapine and olanzapine. Citalopram also showed the lowest occurrence of falls compared to olanzapine, but no difference in lowering falls between citalopram and quetiapine. Citalopram showed lower incidence of orthostatic hypotension than quetiapine and olanzapine.
Pollock et al., 2007 [7]	103 (citalopram n = 53), (risperidone n = 50)	12 weeks	Citalopram, risperidone	N = 31 lost follow-ups in the risperidone group. N = 38 lost follow-ups in the citalopram group	NBRS agitation score citalopram vs. risperidone (OR 0.11 95% CI = −0.28, 0.50 at *p* = 0.57); NBRS psychosis score (OR = 0.06, 95% CI = −0.35, 0.46 at *p* = 0.79); UKU total score (OR = 0.52, 95% CI 0.12, 0.91 at *p* = 0.01); UKU psychotic subscale score (OR = 0.55, 95% CI 0.15, 0.94 at *p* = 0.007); UKU neurological subscale score (OR = 0.22, 95% CI = −0.17, 0.61 at *p* = 0.27).	The result of this trial showed the agitation and psychosis symptoms decreased in both treatment groups. Additionally, the trial stated that there was a significant side effects with risperidone but not with citalopram.
Marano et al., 2014 [6]	186 (citalopram n = 94), (psychotherapy n = 92).	9 weeks	Citalopram vs. placebo	After the 9-week visit, n = 8 lost follow-ups in the citalopram group and n = 9 lost follow-ups in the psychotherapy group.	The NBRS-A scale after 9 weeks shown OR −0.93 95% CI = −1.80, −0.06 at *p* = 0.04. The results of mADCS-CGIC showed that 40% in the citalopram group had marked improvements, with OR 2.13 95% CI = 1.23, 3.69 at *p* = 0.01. QT interval prolongation was seen in the citalopram group and not in the placebo group (18.1 ms; 95% CI = 6.1, 30.1 at *p* = 0.004).	The results of this trial showed significant improvement in the citalopram group compared to the placebo group. Moreover, CMAI and NPI scales revealed significant improvements for citalopram group participants. However, QT intervals prolongation were seen in the citalopram group

**Table 2 pharmacy-10-00061-t002:** AMSTAR Assessment tool.

AMSTAR 2 TOOL	Seitz et al., 2011 [32]	Chaiyakunapruk et al., 2018 [30]
Includes PICO components and research questions	YES	YES
Comprehensive details about methodology before conducting the review	YES	YES
Description of the inclusion criteria	YES	YES
Comprehensive details about searching strategies	YES	YES
Performs study selection in duplicate	YES	NO
Data extraction in duplicate	YES	YES
Describes exclusion criteria	NO	YES
Describes inclusion studies in detail	YES	YES
Satisfactory technique for assessing risk of bias	YES	YES
Report the source of funding	NO	NO
Uses an appropriate statistical technique for meta-analysis combination of RCT results	YES	YES
Assesses the potential impact of RoB on each individual RCT study	NO	YES
Accounts RoB in each individual study when discussing the result	YES	YES
Satisfactory explanation for RCT results in the review	YES	YES
Performs an adequate investigation of potential risk of bias in quantitative studies	YES	YES
Reported any potential conflict of interest and funding	YES	YES

## Data Availability

This study did not report any data.

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
