# Peer review of "A Narrative Review of Studies Comparing Efficacy and Safety of Citalopram with Atypical Antipsychotics for Agitation in Behavioral and Psychological Symptoms of Dementia (BPSD)"

_pharmacy, 2022, doi:10.3390/pharmacy10030061_

Round 1

Reviewer 1 Report

The authors revised the manuscript andi it is now ready for publication.

Author Response

Thanks so much for your support and makes my effort success to the publications 

Reviewer 2 Report

Thank you for the opportunity to review this manuscript. I do believe that this topic is important and awareness of agitation treatment should be raised in scientific literature. 

However, I have several major concerns regarding this manuscript.

  1. this is not a systematic review as you have not included only RCT but also systematic reviews in your review
  2. English language needs to be improved
  3. I am surprised that only two authors conducted this study. Please describe in the methods section are you an expert in this field and what are your qualifications to review this topic
  4. Review articles should include at least 100 references. I suggest that authors include all articles regarding citalopram and agitation (cohort studies, observationals etc). This would be a narrative review where you could include your critical appraisal. 
  5. you have many misspelled words and other typographical errors throughout the manuscript

Before the suggested adjustments I do not find this manuscript to be suitable for publication.

Author Response

The following my answers: 

This is not a systematic review as you have not included only RCT but also systematic reviews in your review

Answer: 

Yes, thanks for your notes. Now it is a Narrative Review (including RCTs, Guidelines, and published systematic reviews) 

English language needs to be improved

Answer: This article already submitted to the English editors in MDPI and right now much better 

I am surprised that only two authors conducted this study. Please describe in the methods section are you an expert in this field and what are your qualifications to review this topic

Answer: 

Me (Haider) and Prof Maree have an extensive experience in academia and in refining and critically appraising clinical research and published articles. 

My qualifications: 

Fellow International College of Neuropsychopharmacology (F.C.I.N.Psych)

Advancing Practice Pharmacist (Adv.P.P.)- Psychiatry & Geriatrics

Graduate Diploma in clinical Practice (including Evidence Based Medicine/ Clinical Epidemiology and Biostatistics) from Monash University. 

GP Pharmacist (Accredited)

Clinical Pharmacist Consultant (Accredited) AACPA

Diabetes and Cardiometabolic Specialist (Sydney Medical School)

Med. Sci., B. Pharm., M.Sc. Medicine (Syd)., AACPA, Dip Mgt., M.R.S (NSW)., M.A.C.C.

Adjunct Senior Lecturer in Pharmacotherapy and Pharmacy Practice /

Charles Sturt University

 Psychiatry Board Certified (Fellowship 4 years in Psychiatry and Geriatrics) 

External Advisory Member in Pharmacy course – CSU/Orange Campus  

Academic Personal Webpage: https://researchoutput.csu.edu.au/en/persons/haider-qasim

ORCID ID: https://orcid.org/0000-0002-7929-6451   

Moreover, the other author is Prof Maree Donna Simpson, is one of the famous leader of clinical research around the world and she is one of the editorial committee of MDPI Pharmacy. 

Review articles should include at least 100 references. I suggest that authors include all articles regarding citalopram and agitation (cohort studies, observationals etc). This would be a narrative review where you could include your critical appraisal.

Answer: We tried me and Prof Maree to find out any updated research in regards out topic. Unfortunately, nothing new. Moreover, I requested from the MDPI committee to give me a month space of time to search and include more studies. But unfortunately nothing new. Also, be remember, this article based on the inclusion and exclusion criteria, and we are strict on them to apply to each article. Also, we assess the validity of the article evidence before we include it into our review. Thus, based on our criteria and honesty applied in this review, we are unable to includes whatever research to reach 100 references. Please be remember, we are believe in quality of what we presenting to the world, NOT the quantity. 

you have many misspelled words and other typographical errors throughout the manuscript 

Answer: This article already submitted to the English editors in MDPI and right now much better 

Round 2

Reviewer 2 Report

the manuscript is appropriate for publication

This manuscript is a resubmission of an earlier submission. The following is a list of the peer review reports and author responses from that submission.

Round 1

Reviewer 1 Report

This review provides an interesting overview of the importance of pharmacological interventions for agitation by evaluating evidence from RCTs, systematic reviews and meta-analyses on BPSD patients. Although the study is well performed, I have two major considerations:

-The reference section is too poor. Authors should include more studies, I provide some studies below.

-The introduction is not sufficient. Authors should include:

1) a brefly introduction on the impact of dementia and on BPSD (current classification, phenomenology, prevalence and incidence, all symptoms and assessment) and cited for example these papers:

  • Ferri CP, Prince M, Brayne C, Brodaty H, Fratiglioni L, Ganguli M, et al. Global prevalence of dementia: a Delphi consensus study. Lancet. 2005 Dec 17;366(9503):2112–7.
  • Aalten P, de Vugt ME, Lousberg R, Korten E, Jaspers N, Senden B, et al. Behavioral problems in dementia: a factor analysis of the neuropsychiatric inventory. Dement Geriatr Cogn Disord. 2003;15(2):99–105.
  • Desai AK, Grossberg GT. Recognition and Management of Behavioral Disturbances in Dementia. Prim Care Companion J Clin Psychiatry. 2001 Jun [cited 2014 Jul 7];3(3):93–109.
  • Ballard C, Howard R. Neuroleptic drugs in dementia: benefits and harm. Nat Rev Neurosci. 2006 Jun;7(6):492–500.
  • Lawlor BA. Behavioral and psychological symptoms in dementia: the role of atypical antipsychotics. J Clin Psychiatry. 2004 Jan;65 Suppl 1:5–10.
  • Overshott R, Byrne J, Burns A. Nonpharmacological and pharmacological interventions for symptoms in Alzheimer’s disease. Expert Rev Neurother. 2004 Sep;4(5):809–21.
  • Boeve BF, Silber MH, Ferman TJ. Current management of sleep disturbances in dementia. Curr Neurol Neurosci Rep. 2002 Mar;2(2):169–77.
  • Savva GM, Zaccai J, Matthews FE, Davidson JE, McKeith I, Brayne C. Prevalence, correlates and course of behavioural and psychological symptoms of dementia in the population. Br J Psychiatry. 2009;194(3):212–9.
  • Finkel S. Introduction to behavioural and psychological symptoms of dementia (BPSD). International journal of geriatric psychiatry. 2000.

-The management of BPSD should be comprehensive of some references on non-pharmacological approach (only cited in line 33-34) as:

  • O’Neil M. E.et al. 2011. A Systematic Evidence Review of Non-pharmacological Interventions for Behavioral Symptoms of Dementia. Washington, DC: Department of Veterans Affairs
  • Kong EH, et al. Aging Ment Health. 2009 Jul; 13(4):512-20.
  • Livingston G, et al. Am J Psychiatry. 2005 Nov; 162(11):1996-2021.
  • Lai CK,et al. Cochrane Database Syst Rev. 2009 Oct 7; (4):CD006470.
  • Iwata B. et al 1993. in Perspectives on the Use of Nonaversive and Aversive Interventions for Persons with Developmental Disabilities, eds Repp A. C., Singh N. N. (Sycamore, IL: Sycamore Press;), 301–330
  • Salzman C, et al J Clin Psychiatry. 2008 Jun; 69(6):889-98.
  • Choi AN, et al. Int J Neurosci. 2009; 119(4):471-81.
  • Ballard C, et al. Am J Geriatr Psychiatry. 2009 Sep; 17(9):726-33.
  • Weitzel T, et al. J Nurses Staff Dev. 2011 Sep-Oct; 27(5):220-6.
  • Mossello E, et al Int Psychogeriatr. 2011 Aug; 23(6):899-905.
  •  

- Please double check spelling and editing, I found several error as:

Line 111-112: should “;” be removed from the sentence?

Line 244 and 249: please, check and remove “.” from the sentences.

Reviewer 2 Report

The authors performed a qualitative review to compare the efficacy and safety of citalopram with atypical antipsychotics on agitation in patients with BPSD. The rationale to choose citalopram as the principal medication to compare with atypical antipsychotics is not been clarified. The search strategy that includes ‘olanzapine’, ‘haloperidol’, ‘risperidone’, ‘quetiapine’ is not accurate if the authors aimed to compare citalopram with “atypical antipsychotics”. If the authors intended to compare the effectiveness of antidepressants with antipsychotics, as they mentioned in the introduction, they should also include other classes of antidepressants and antipsychotics. This renders the findings very ambiguous. The subject is interesting and important to the field, but the presentation has currently no benefit to the readers.